# Histology Agnostic Drug Development: An Updated Review

**DOI:** 10.3390/cancers16213642

**Published:** 2024-10-29

**Authors:** Kevin Nguyen, Karina Fama, Guadalupe Mercado, Yin Myat, Kyaw Thein

**Affiliations:** 1Touro University Nevada College of Osteopathic Medicine, 874 American Pacific Dr, Henderson, NV 89014, USA; knguyen49@student.touro.edu (K.N.);; 2University College Dublin School of Medicine, Belfield, D04 V1W8 Dublin, Ireland; 3One Brooklyn Health—Interfaith Medical Center Campus, 1545, Atlantic Avenue, Brooklyn, NY 11213, USA; 4Comprehensive Cancer Centers of Nevada—Central Valley, 3730 S Eastern Ave, Las Vegas, NV 89169, USA; 5Kirk Kerkorian School of Medicine, University of Nevada, Las Vegas (UNLV), 4505 S. Maryland Pkwy, Las Vegas, NV 89154, USA

**Keywords:** tissue agnostic drug development, microsatellite instability-high (*MSI-H*), tumor mutational burden-high (*TMB-H*), neurotrophic tyrosine kinase (*NTRK*), V-raf murine sarcoma viral oncogene homolog B *V600E* (*BRAF^V600E^*), rearranged during transfection (*RET*)

## Abstract

Cancer treatments traditionally focus on the type of tissue where the cancer starts, but recent research has shifted to targeting genetic changes that drive cancer growth, regardless of where the tumor originates. This approach is known as histology-agnostic therapy. In this review, we provide an update on these therapies, focusing on several newly developed drugs that are changing the way cancer is treated. Our goal was to explain how these drugs work, the benefits they offer, and the challenges that remain, such as resistance to treatment. This research may help doctors and researchers better understand how to choose the right treatments for patients with different types of cancer. By addressing the strengths and limitations of histology-agnostic therapies, we hope to support ongoing efforts in cancer treatment and encourage further studies to improve outcomes for patients.

## 1. Introduction

Over the last decade, the field of cancer treatment has witnessed a transformative shift with the emergence of tumor-agnostic therapies (Figure 1). Traditionally, cancer treatments were prescribed based on histological type; however, the introduction of basket trials marks a significant departure from this approach. These trials evaluate the effectiveness of drugs against specific mutations or biomarkers present across diverse tumor types, irrespective of the tissue of origin. This methodology is particularly advantageous as it enables the inclusion of a broader array of tumor types in clinical studies, benefiting patients who might not otherwise qualify for targeted therapies. Moreover, basket trials facilitate a more rapid assessment of therapeutics by simultaneously collecting extensive data on the safety and efficacy of a drug across various forms of cancer. This approach not only streamlines the drug evaluation process but also enhances our understanding of oncological pharmacotherapy, paving the way for more personalized and effective treatment regimens.

A landmark in this journey was the accelerated endorsement of pembrolizumab for targeting solid tumors marked by high microsatellite instability (*MSI-H*) or deficient mismatch repair (*dMMR*), setting a precedent for future therapeutic innovations (Figure 1). In March 2023, this treatment achieved full endorsement based on its genetic efficacy, marking a significant milestone. Following this, larotrectinib also secured approval for the treatment of solid tumors in both adults and children, specifically those with neurotrophic tyrosine kinase (*NTRK*) fusion positivity. Notably, recent evaluations of larotrectinib have shown its particular effectiveness against salivary gland tumors, further broadening the spectrum of targeted cancer treatments.

Previous to this, several other clinical trials have successfully introduced *NTRK* inhibition as an ideal target for multiple cancer processes. In August of 2019, entrectinib gained FDA approval for *NTRK*-fusion positive (*NTRK-fp*) tumors such as non-small cell lung carcinoma (NSCLC). Continued research into the improvement of this drug followed to target common solid tumors with known *NTRK* involvement, such as colorectal cancer, gynecologic cancers, and melanoma. Dostarlimab was granted accelerated approval by the FDA in April 2021 for *dMMR* recurrent advanced endometrial cancer that continues to progress following a platinum-containing regimen, which subsequently was approved in February 2023 for the same cancer type, this time including recurrent or advanced endometrial cancer that has progressed following a platinum-containing regimen, curative surgery, or radiation.

Furthermore, selpercatinib was approved by the FDA on 21 September 2022 for the treatment of locally advanced or metastatic solid tumors that demonstrate *RET* gene fusion (Figure 2). This agnostic therapy has been approved in adult patients whose solid tumors progressed on or after prior systemic treatment or patients who do not have alternative successful treatments. The target of this drug is the mutated isoforms of *RET*, but it contains activity toward other genes such as *VEGF1* and *VEGFR3*. Pembrolizumab was approved on 16 June 2020 and treats a wide age range of patients with unresectable or metastatic solid tumors with a high tumor mutational burden, as this has been theorized to be responsive to immunotherapy. Lastly, dabrafenib, in combination with trametinib, was approved on 22 June 2022 in adult or pediatric patients with unresectable or metastatic solid tumors with the *BRAF^V600E^* mutation in adults or pediatric patients, with the exception of patients with colorectal cancer. More recently, advancements in tumor-agnostic treatments came with the approval of trastuzumab deruxtecan-nxki (T-Dxd) by the FDA on 5 April 2024, for unresectable or metastatic *HER2*-positive solid tumors and reprotrectinib on 13 June 2024 for NTRK-fp solid tumors (Figure 2). We aim to elucidate further the data associated with these tumor-agnostic therapies and discuss any further advancements in the field.

## 2. Promises and Pitfalls of Histology-Agnostic Therapy

Histology-agnostic therapies have revolutionized cancer treatment by targeting specific molecular alterations irrespective of the tumor’s tissue of origin. These therapies, such as larotrectinib and entrectinib for NTRK fusions and pembrolizumab for tumors with MSI-H, have shown promising results in a variety of malignancies, demonstrating the potential for broad application. However, the clinical utility of these treatments is limited by several key challenges. Heterogeneous responses across different tumor types remain a major issue, as exemplified by the diminished efficacy of BRAF inhibitors in colorectal cancer compared with melanoma due to feedback activation of alternative pathways. Resistance mechanisms, including secondary mutations, gene amplifications, and adaptive pathway reactivation, further complicate treatment and can lead to therapeutic failure over time. Understanding the various pharmacologic mechanisms of action is also a key aspect of understanding the efficacy and resistance of these treatments (Figure 3). Additionally, effective implementation requires advanced diagnostic testing, such as comprehensive genomic profiling, which may not be widely available, particularly in low-resource settings. Patient stratification is another hurdle, as identifying the small subset of patients harboring these rare mutations can complicate trial recruitment and reduce the statistical power of studies. Addressing these barriers through improved trial designs, a better understanding of resistance mechanisms, and the development of standardized diagnostic criteria is essential to maximize the impact of histology-agnostic therapies and broaden their clinical application [1].

I.Approval of pembrolizumab in patients with unresectable or metastatic *dMMR*/*MSI-H* cancers (May 2017)

In a landmark development, pembrolizumab was granted accelerated approval in May 2017 as the first tumor-agnostic therapy, showcasing notable efficacy in patients with *MSI-H*/*dMMR* tumors (Figure 2). This approval was predicated on findings from five multicohort, single-arm clinical trials—KEYNOTE-016, -164, -012, -028, and -158—conducted by Merck. These studies evaluated 149 patients with a variety of cancer types, with a significant focus on colorectal cancer, which constituted 60% of cases (Table 1). The determination of *MSI-H* or *dMMR* tumor status in the majority of these patients (135/149) was achieved prospectively via PCR tests for *MSI-H* status or IHC tests for *dMMR* in local laboratories. The trials revealed an overall response rate (ORR) of 36% in colorectal cancer patients and 46% in those with other tumor types [2].

Pembrolizumab subsequently earned full approval as the inaugural immunotherapy based on the outcomes of phase 2 trials KEYNOTE-158, -164, and -051. KEYNOTE-164 assessed 124 patients with advanced *MSI-H*/*dMMR* colorectal cancer following progression after fluoropyrimidine-based and oxaliplatin or irinotecan therapies, with or without anti-VEGF/EGFR mAb-based treatments [12]. KEYNOTE-158 studied 373 patients with advanced *MSI-H*/*dMMR* non-colorectal cancers post-therapy progression, prospectively enrolling *MSI-H/dMMR* tumor patients in cohort K [13]. KEYNOTE-051, meanwhile, focused on 7 pediatric patients with *MSI-H/dMMR* cancers [14]. The primary endpoints of ORR and DOR were evaluated through a blinded independent central review in KEYNOTE-158 and by investigators in KEYNOTE-051 according to RECIST v1.1 criteria. 

An integrated analysis of these trials reported an ORR of 33.3% (95% CI: 29.2, 37.6), with a complete response rate (CRR) of 10.3% and a partial response rate of 23.0% across a median follow-up period of 20.1 months. Notably, 77% of the 168 responding patients had responses lasting at least 12 months, with 39% extending beyond 36 months. The median duration of response (DOR) was recorded at 63.2 months. Specifically, an ORR of 34% (95% CI: 26, 43) and a DOR ranging from 4.4 to 58.5+ months were observed in *MSI-H/dMMR* colorectal cancer patients (*n* = 124). For non-colorectal solid tumor patients with *MSI-H/dMMR* (*n* = 380), pembrolizumab achieved an ORR of 33% (95% CI: 28, 38) and a similarly extended DOR [15]. The therapy’s safety profile was characterized by manageable adverse effects, including pruritus, fatigue, diarrhea, and others, reinforcing pembrolizumab’s pivotal role in the precision oncology landscape for MSI-H/dMMR cancers.

II.Approval of larotrectinib in patients with unresectable or metastatic *NTRK* gene fusion-positive cancers (November 2018)

While neurotrophic tropomyosin kinase receptors (*NTRK*) gene fusions are increasingly recognized as predictive biomarkers, their prevalence varies significantly between rare and common cancers. These fusions are highly prevalent in certain rare cancers, found in 90–100% of mammary analog secretory carcinoma of the breast, over 90% of mammary secretory carcinoma, 91–100% of fibrosarcoma, and 83% of congenital mesoblastic nephroma. In contrast, *NTRK* fusions are much less common in widespread cancers, occurring in less than 5%. Specific rates include about 0.2% in head and neck neoplasms, 0.2–0.3% in pulmonary cancer, 0.7–1.5% in colorectal cancer, 0.3% in cutaneous melanoma, and 1% in sarcoma [16]. This contrast highlights the importance of targeted genomic screening, particularly in common cancers where *NTRK* fusions are rare.

Larotrectinib, a groundbreaking CNS-active inhibitor targeting *TRKA*, *TRKB*, and *TRKC*, secured accelerated approval in 2018 for the treatment of *TRK* fusion cancer (Figure 2). This seminal decision was predicated on compelling evidence from three pivotal phase I/II trials: LOXO-TRK-14001 (NCT02122913), NAVIGATE (NCT02576431), and SCOUT (NCT02637687). In an unprecedented pooled analysis, Hong and colleagues [17] demonstrated an ORR of 79% and a complete response (CR) rate of 16% among participants recruited from May 2014 through February 2019 (Table 1). Eligibility for this analysis was meticulously defined, requiring participants to be at least one month old, with a diagnosis of locally advanced or metastatic non-CNS primary solid tumors harboring *TRK* fusions, and to have undergone any accessible standard therapies [17].

Salivary gland tumors, non-gastrointestinal stromal tumor soft-tissue sarcomas, infantile fibrosarcoma, and thyroid cancer were the predominant tumor types, encompassing 20%, 20%, 13%, and 9% of the cohort, respectively. Remarkably, the aggregated data from 260 patients treated irrespective of *TRK* fusion status revealed no incidences of treatment-related mortality. The analysis identified the most common severe adverse events as increases in alanine aminotransferase and aspartate aminotransferase, as well as anemia and neutrophil count reduction, reflecting a manageable safety profile [17].

Since its landmark approval, larotrectinib has consistently demonstrated remarkable efficacy. Hong and colleagues [18] presented updated findings from an expanded cohort across the initial three pivotal clinical trials. In these studies, most patients received larotrectinib at a dosage of 100 mg twice daily, with responses evaluated by an independent review committee. The cohort included 180 patients representing 24 distinct tumor types, notably lung (15%), soft tissue sarcoma (15%), thyroid (14%), salivary gland (13%), and colon (12%). The ORR stood at 57% (95% CI: 50, 65), with 16% achieving complete responses (including one pathological complete response), 41% partial responses, 22% stable disease, 13% progressive disease, and 8% not evaluable. Among the 22 patients with known brain metastases at baseline, the ORR reached 68%. Median overall survival was observed at 48.7 months. Treatment-related adverse events (TRAEs) were primarily Grade 1/2, and only one patient discontinued treatment due to elevated levels of alanine aminotransferase and aspartate aminotransferase.

This broader dataset underlines the sustained effectiveness and favorable safety profile of larotrectinib, reinforcing its therapeutic value across a diverse spectrum of tumor types.

III.Approval of entrectinib in patients with unresectable or metastatic *NTRK* gene fusion-positive cancers (August 2019)

Entrectinib is an *NTRK* inhibitor that is designed to penetrate the blood–brain barrier to target *NTRK* gene tumors without a known acquired resistance mutation, are metastatic or where surgical resection is likely to result in severe morbidity and has progressed following treatment or have no satisfactory alternative therapy [19]. It is a multi-kinase inhibitor that features a metabolite, designated M5, which serves as an inhibitor of the tropomyosin receptor tyrosine kinase *NTRK*, proto-oncogene tyrosine kinase, *ROS1*, as well as anaplastic lymphoma kinase (*ALK*). In comparison to other *NTRK* drugs, entrectinib has demonstrated deep and durable systemic and IC responses in patients with *NTRK-fp* tumors, including NSCLC. The drug was approved by the FDA on 15 August 2019 (Figure 2), based on an ORR of 57% yielded from three multicohort clinical trials, ALKA-372-001, STARTRK-1, and STARTRK-2, which includes a 7% CRR, with 68% of the responses lasting ≥ 6 months (Table 1). The most common cancer types included lung (56%), sarcoma (8%), and colon (5%) [20]. Some notable adverse effects of the drug were pulmonary infections, weight gain, dyspnea, fatigue/asthenia, cognitive disorders, syncope, pulmonary embolism, hypoxia, pleural effusion, hypotension, diarrhea, and urinary tract infections [19]. Serious risks included congestive heart failure (CHF), CNS adverse reactions, skeletal fractures, hyperuricemia, hepatotoxicity, QT prolongation, and vision disorders [19].

Uncertain risks were posed following the approval of entrectinib due to the limited generalizability of the results of the trials. It was suggested that to address these concerns, data from additional patients with more common solid tumors that contain *NTRK* fusions, such as colorectal cancer, gynecologic cancers, and melanoma, as well as the inclusion of pediatric patients with *NTRK* fusion solid tumors and primary CNS malignancies, can be helpful. In a most recent integrated analysis, entrectinib showed a durable intracranial response in 11 patients with measurable CNS disease, further proving its potential in being a CNS active treatment in patients with *NTRK* fusion-positive solid tumors [21].

IV.Approval of pembrolizumab approved for patients with unresectable or metastatic *TMB-H* cancers (16 June 2020)

Pembrolizumab received FDA approval on 16 June 2020 for treating adult and pediatric patients with unresectable or metastatic solid tumors with a tumor mutational burden high (*TMB-H*) [2]. Tumor mutational burden (TMB) indicates the somatic genomic burden of mutations within a given tumor, and a higher mutational burden increases neoantigen formation [22]. Various reports have shown that a high TMB leads to neoantigen formation with possible immune system recognition, meaning that the tumor will be responsive to immunotherapy [2]. The currently accepted threshold for *TMB-H* is ≥10 mut/Mb, as this indicates an increased likelihood of neoantigen formation [23], and pre-specified cutoff points of ≥10 and ≥13 mut/Mb using the FoundationOne CDx assay were used in the clinical trial [22]. TMB varies widely between malignancies and the assays in evaluating TMB [22]. The clinical trial KEYNOTE-158 included patient tumor response data where TMB was derived from whole exome sequencing [22].

The efficacy of pembrolizumab in *TMB-H* advanced malignancies following prior treatment or no satisfactory alternative treatment was evaluated in KEYNOTE-15, an open-label, nonrandomized, multicenter, multicohort trial of pembrolizumab [2]. In this clinical trial, patients received pembrolizumab 200 mg every three weeks until they had unacceptable toxicity or disease progression. In this trial, 1,050 patients received at least one dose of pembrolizumab [22]. In the trial, 260 patients lacked TMB scores for various reasons, and from the remaining 790 patients, only 13% (N = 102) met the criteria for *TMB-H* [22]. ORR in treating pembrolizumab was 29.4% in patients with a *TMB-H* ≥ 10 mut/Mb, with only 4% of patients experiencing a complete response rate (Table 1) [24]. Additionally, *TMB-H* ≥ 13 mut/Mb patients had an ORR of 37%, with 3% experiencing a complete response rate [24]. Amongst the 102 *TMB-H* patients, ORR varied by tumor type, indicating that further investigation is needed to determine appropriate *TMB-H* thresholds for different types of malignancies if the *TMB-H* ≥ 10 mut/Mb is not sufficient [22].

V.Approval of dostarlimab in patients with unresectable or metastatic *dMMR* cancers (August 2021)

Dostarlimab is an IgG4 humanized anti-PD1 monoclonal antibody derived from a mouse hybridoma. It belongs to a classification of cancer drugs known as immune checkpoint blockers (ICBs), which target the immunosuppressive activity of cancer cells, reviving the host’s immune response to the tumor [25]. Targeting the PD-1/PD-L1 axis, this drug inhibits T-cell proliferation, apoptosis, and exhaustion (Figure 3). The safety and activity of dostarlimab were assessed across multiple solid tumors in the open-label, single-arm multicohort phase I GARNET trial (NCT02715284). In this trial, dostarlimab showed meaningful activity in *dMMR*/*MSI* endometrial cancer with an ORR of 42.3% (95% CI 30.6, 54.6) with a CR of 12.7% and a partial response (PR) of 29.6% [25]. The resulting most frequent treatment-related adverse events (TRAEs) were reported in 10% of the patients involved. These TRAEs included asthenia, diarrhea, fatigue, and nausea. Dostarlimab also demonstrated outstanding effectiveness in locally advanced rectal cancer (LARC). It provided a way to subject the cancer cells to an immune blockage at an earlier stage. In the phase II NCT04165772 trial for the drug, all 12 patients enrolled in the study achieved 100% CR, with five out of the 12 patients showing signs of early clinical CR [25]. However, considering the small sample size, additional data should be considered for further evidence of its effectiveness. Of note were certain adverse events, including rash, pruritus, and fatigue.

In the GARNET trial cohort involving patients with *EGFR* or *ALK*-mutated NSCLC, the ORR was 26.9%, with two patients achieving CR [25]. The TRAEs included fatigue, hypothyroidism, and asthenia. Currently, dostarlimab is being evaluated as monotherapy or part of combination therapy for gynecological malignancies. The drug is currently being incorporated in several trials, including NCT039817896, wherein its effect in advanced, treatment-naïve endometrial cancer. Phase II of the GARNET trial is attempting to study the safety and efficacy of dostarlimab in high-risk locally advanced cervical cancer. The OPAL trial failed to show any benefit of dostarlimab in conjunction with niraparib in therapy for patients with platinum-resistant ovarian cancer and *BRCA* wild-type tumors [25]. This combination regimen is now being evaluated for other solid tumors such as head and neck squamous cell carcinoma, penile cancer, and small cell lung cancer. The GARNET trial was instrumental in the FDA approval of dostarlimab for all types of solid tumors harboring mismatch repair. The results of phase II and phase III of this trial will determine if dostarlimab’s activity can extend to *d-MMR* non-endometrial cancers.

VI.Approval of dabrafenib plus trametinib approved for patients with unresectable or metastatic *BRAF^V600E^* positive cancers (June 2022)

Dabrafenib, in combination with trametinib, received accelerated FDA approval on 22 June 2022 for the treatment of unresectable or metastatic solid tumors with the *BRAF^V600E^* mutation in adults or pediatric patients (≥6 years) (Figure 2) [8]. It is indicated in patients whose tumors have progressed following prior treatment or have no other alternative treatment options except those with colorectal cancer due to resistance to *BRAF* inhibition [8]. B-Raf proto-oncogene (*BRAF*) is a serine-threonine kinase that can activate the *MAPK* pathway (Figure 3) [26]. These mutations are detected in 7–15% of all cancers, including rare cancers such as gliomas, biliary tract cancer, gastrointestinal stromal tumors, etc. [26]. The mechanism of action of dabrafenib is that it selectively inhibits the mutated *BRAF* kinase. However, a major drawback of dabrafenib monotherapy is that most patients will progress, limiting the clinical benefit [27]. Studies have shown that using *BRAF* inhibitors can paradoxically lead to the activation of the *MAPK* pathway, and this is associated with the progression of the cancer [27]. At the same time, trametinib functions via selective, reversible inhibition of *MEK1* and *MEK2* activation, and therefore, the use of both agents increases the anti-tumor activity and has become the standard of care [26].

The effectiveness of dabrafenib plus trametinib was assessed in three distinct trials (ROAR, NCI-MATCH, and Study X2101) involving patients diagnosed with various types of cancer that contained the *BRAF^V600E^* mutation [27]. In the ROAR trial (BRF117019), the cancer diagnoses included high-grade glioma, low-grade glioma, biliary tract cancer, adenocarcinoma of the small intestine, gastrointestinal stromal tumor, and anaplastic thyroid cancer [8]. In comparison, the NCI-MATCH trial (EAY121-H) included any adults with *BRAF^V600E^* mutation-positive tumors, such as gastrointestinal tumors, lung tumors, gynecological or peritoneal tumors, CNS tumors, and ameloblastoma of the mandible. However, it excluded patients with melanoma, thyroid cancer, or colorectal cancer [27]. In the adult trials, there were 131 enrolled patients with 24 different tumor types, and 54 experienced an objective response [8]. The ORR was 46% (95% CI: 31, 61) in biliary tract cancer, 33% (95% CI: 20, 48) for high-grade glioma, and 50% (95% CI: 23, 77) for low-grade glioma [8]. The most frequently reported adverse reactions occurring in over 20% of the patients included pyrexia, fatigue, nausea, rash, chills, headache, hemorrhage, cough, vomiting, constipation, diarrhea, myalgia, arthralgia, and edema [8].

Study X2101 investigated pediatric patients with recurrent or refractory low-grade and high-grade glioma [8]. In the pediatric trial, the ORR was 25% (95% CI: 12, 42) with a DOR of ≥6 months for 78% of patients and ≥24 months for 44% of patients. The most frequently recorded adverse reactions in the pediatric category occurring in 20% of patients include pyrexia, rash, vomiting, fatigue, dry skin, cough, diarrhea, dermatitis, acneiform, headache, abdominal pain, nausea, hemorrhage, constipation, and paronychia [8].

VII.Approval of selpercatinib in patients with unresectable or metastatic *RET*-positive cancers in patients ≥ 12 years (September 2022)

Selpercatinib gained FDA approval on 21 September 2022 for treating adult patients with locally advanced or metastatic solid tumors that demonstrate *RET* gene fusion (Figure 2) [28]. This drug is indicated in patients whose solid tumors progressed on or after prior systemic treatment or who do not have alternative treatment options [28]. Promising early efficacy data in LIBRETTO-001 alongside the efficacy data demonstrated in the 343 patients with *RET* fusion-positive NSCLC and thyroid cancer enrolled in the same trial allowed the drug to gain the accelerated FDA approval designation. Of note, selpercatinib gained prior approval on 8 May 2020 for the treatment of metastatic *RET* fusion-positive NSCLC in adults and advanced or metastatic *RET* mutant medullary thyroid cancer in patients 12 years of age or greater [29]. Selpercatinib is a kinase inhibitor that inhibits both wild-type and mutated isoforms of *RET*, as well as other genes, such as *VEGFR1* and *VEGFR3* [29]. Alterations such as fusions or point mutations in *RET* increase oncogenic potential and are most commonly found in NSCLC and papillary thyroid cancer. *CCDC6-RET*, *KIF5B- RET*, *RET V804M*, and *RET M918T* are all gene fusions and mutations that are responsive to the anti-tumor activity of selpercatinib [30].

The efficacy of selpercatinib was evaluated in an ongoing multicenter, open-label, multicohort trial, LIBRETTO-001 (NCT03157128) [28]. This trial evaluated 41 patients with *RET* fusion-positive tumors beyond NSCLC and thyroid cancer (Figure 1) [28]. The ORR was 44% (95% CI: 28, 60) with a DOR of 24.5 months (95% CI: 9.2, NE) [28]. Additionally, 67% (95% CI: 41, 87) of patients were still responding positively at 6 months, and 56% (95% CI: 31, 78) were responding at 12 months (Table 1) [28]. Among the 41 patients, there were varied tumor types that demonstrated response to selpercatinib, including pancreatic adenocarcinoma, colorectal, salivary, unknown primary, breast, soft tissue sarcoma, bronchial carcinoid, ovarian, small intestine, and cholangiocarcinoma [28]. Approximately 90% (n = 37) of the patients included in this study had previously undergone systemic therapy, with a median number of treatments being 2, and 32% had received three or more treatments [30]. The safety and adverse events information included data from all patients (n = 756) treated with selpercatinib in LIBRETTO-001 [30]. The most frequently reported adverse reactions, with occurrence rates exceeding 25% among patients, were edema, diarrhea, fatigue, dry mouth, hypertension, abdominal pain, constipation, rash, nausea, and headache [28].

A recent update in the ongoing study includes more patients (n = 52) and 16 months longer follow-up being treated with selpercatinib for *RET*-activated cancers [31]. The majority of the patients (n = 31) had GI *RET*-positive cancers, with pancreatic and colorectal being the most common [31]. In this group of patients, the ORR was 44.2% (95% CI: 30.5, 58.7) and varied between cancer types, with the ORR in pancreatic tumors being 53.8% and 30.8% in colorectal tumors [31]. The median DOR across all tumor types was 37.2 months [31]. These updated data continue to highlight the anti-tumor activity of selpercatinib against *RET*-positive tumors.

VIII.Approval of trastuzumab deruxtecan approved for patients with unresectable or metastatic *HER2*-positive cancers (April 2024)

The amplification or overexpression of the Human Epidermal Growth Factor Receptor 2 (*HER2*) has been implicated in the pathogenesis of a diverse array of malignancies, encompassing breast, bladder, pancreatic, gallbladder, and esophageal cancers [32]. Recent advancements in oncologic therapeutics have heralded the accelerated approval of fam-trastuzumab deruxtecan-nxki for the management of adults with unresectable or metastatic solid tumors characterized by *HER2* overexpression (IHC3+). The immunohistochemistry (IHC) scoring system assesses *HER2* expression on the surface of tumor cells, which is crucial for determining treatment eligibility. In this system, a score of IHC1+ signifies no significant *HER2* expression; IHC2+ is considered equivocal and may require further testing, while IHC3+ confirms strong *HER2* positivity, indicating that the tumor may respond well to therapies targeting the HER2 protein. This therapeutic agent embodies a novel conjugated antibody-drug amalgamating trastuzumab—a monoclonal antibody with high affinity for the HER2 receptor—with a potent cytotoxic moiety, deruxtecan, facilitating the targeted delivery of the cytotoxic agent directly to the cancer cells (Figure 2).

The basis for this approval was furnished by the outcomes from three key multicenter trials: DESTINY-PanTumor02 (NCT04482309), DESTINY-Lung01 (NCT03505710), and DESTINY-CRC02 (NCT04744831). In DESTINY-PanTumor02, the observed ORR was 51.4%, with a median duration of response (DOR) extending to 19.4 months [33]. Similarly, DESTINY-Lung01 reported an ORR of 52.9%, with a median DOR of 6.9 months [34], while DESTINY-CRC02 demonstrated an ORR of 46.9%, with a DOR of 5.5 months (Table 1) [35]. Notably, subjects with a precedent of interstitial lung disease (ILD) were excluded from participation in these studies.

The therapeutic regimen was associated with a spectrum of adverse reactions and laboratory abnormalities in no less than 20% of participants, including but not limited to hematological changes (e.g., reductions in white blood cells, hemoglobin, neutrophils, lymphocytes, and platelets) and enzymatic elevations (aspartate aminotransferase, alanine aminotransferase, blood alkaline phosphatase), alongside clinical manifestations such as nausea, fatigue, vomiting, decreased appetite, alopecia, diarrhea, electrolyte imbalances, constipation, stomatitis, and upper respiratory tract infections. Of note is that respiratory toxicity due to T-Dxd was significant. Treatment-related ILD occurred in 26% of patients and resulted in death in two patients in DESTINY-Lung01 [36]. Despite its accelerated endorsement for this tumor-agnostic indication, the imperative for continued validation of fam-trastuzumab deruxtecan-nxki’s clinical utility through confirmatory trials remains paramount.

IX.Approval of repotrectinib approved for NTRK-positive tumors (June 2024)

The American Association for Cancer Research (AACR) recently announced the FDA’s accelerated approval of repotrectinib on13 June 2024 for the treatment of solid tumors with *NTRK* mutations in adults and pediatric patients aged 12 years and older (Figure 2). Repotrectinib was first approved based on the TRIDENT-1 trials in November 2023 for the initial treatment of *ROS1*-linked NSCLC or as a second-line treatment in patients who have previously received *ROS1*-targeted drugs [37] such as larotrectinib. This same clinical trial has now resulted in the drug’s designated approval for locally advanced or metastatic tumors that have been resistant to treatment or have shown no adequate alternative treatment.

Tyrosine kinase inhibitors (TKIs) have demonstrated efficacy in treating such cancers, but resistance invariably develops. Specifically, mutations at the kinase solvent front, known as solvent front mutations (SFMs), can lead to resistance. SFMs affect a conserved amino acid sequence near the ATP-binding site in kinases. Examples of these mutations include ALKG1202R in ALK-rearranged tumors, ROS1G2032R and ROS1D2033N in ROS1-rearranged tumors, and TRKAG595R and TRKCG623R in NTRK1- and NTRK3-rearranged tumors. Repotrectinib has shown potent inhibitory activity against these clinically challenging SFMs, as well as wild-type (WT) ROS1, TRKA–C, ALK, and other relevant non-SFM mutations [38].

The TRIDENT-1 trial, which included 88 adult patients, reported a significant ORR of 58% in TKI-naïve patients and 50% in those previously treated with a TKI, with a prolonged DOR in TKI-naïve patients (Table 1). Adverse reactions observed in more than 20% of patients included dizziness, dysgeusia, peripheral neuropathy, constipation, dyspnea, fatigue, ataxia, cognitive impairment, muscular weakness, and nausea [11]. Additionally, an ongoing first-in-human dose-escalation trial demonstrated the anti-tumor activity of repotrectinib in patients with ROS1- or NTRK3-rearranged tumors harboring resistant SFMs.

Although safety, dosing, and clinical efficacy are still being established, these findings indicate that repotrectinib could represent an effective treatment option for *ROS1/NTRK*-rearranged malignancies, including those with resistant SFMs.

## 3. Challenges, Remaining Questions and Future Directions

The landscape of tumor-agnostic drug development has experienced a significant revolution over recent years, characterized by numerous advanced regulatory approvals, including the landmark full approval of Pembrolizumab for *MSI-H* or *dMM* malignancies (Figure 1). As more studies gain approval and proceed, the long-term efficacy and safety profiles of these innovative treatments will become clearer, further informing their clinical use. Currently, these therapies are emerging as preferred options for the management of refractory cancers, demonstrating notable effectiveness in terms of ORR and treatment-emergent adverse events (TAEs). Nonetheless, the clinical application of these drugs is sometimes hindered by uncertainties, such as the limited availability of patients with specific tumor types or the potential incongruence between the mutation prevalence in clinical trials and that observed globally.

(a)Does one size fit all?

The approved tumor-agnostic therapies have expanded the use of these medications for patients who otherwise have unmet clinical needs. However, one limitation is their efficacy across different cancer types. This is likely due to multiple interlinking factors related to genomic heterogeneity, resistance, and immune escape.

Tumor heterogeneity can be defined as the diverse mutations found in populations and/or subpopulations between different tumor cells (intertumor heterogeneity) or within the same tumor cell (intratumor heterogeneity) [39]. For instance, CRC presents a unique challenge due to its heterogeneous nature. Certain genetic and molecular subtypes specific to CRC may affect how it will behave and respond to therapy. As previously discussed, there are cancer-specific differences in the efficacy of the agnostic drugs, and this can be attributed to resistance. This is highlighted by the drug combination of dabrafenib plus trametinib, which is indicated in unresectable or metastatic solid tumors with a *BRAF^V600E^* mutation, except in colorectal cancer, due to intrinsic resistance to *BRAF* inhibition [8]. While intrinsic resistance to *BRAF/MEK* inhibitors is rare, acquired resistance is widespread and usually follows treatment [40]. There are multiple genes, such as *KRAS*, *NRAS*, *MAP2K1*, and *MAP2K2*, as well as *BRAF* amplifications, that can lead to resistance by reactivating the *MAPK* pathway in the presence of *BRAK/MEK* inhibitors [40]. Other resistance mechanisms include splice variants of *RAF^V600E^*, mutations in various genes in the *PIK2CA/PTEN* signaling pathway, and mutations in *RAC1*, *CDK4*, *CCND1*, and c-*MET*. In some cases where there is no clear genetic cause of resistance, epigenetic changes are suspected to be responsible for resistance. Additionally, recent literature studies suggest a role in androgen receptor expression as a potential mechanism for resistance in animal models [41]. Male mice expressed significantly higher androgen receptors and had impaired anti-tumor activity when treating melanoma [41]. These results were then further confirmed in a cohort of melanoma patients treated with *BRAF*/*MEK* inhibitors [40], indicating a possible role for concurrent androgen suppression in treating patients. The complex influence of multiple genes on the efficacy of *BRAK*/*MEK* inhibitors poses a significant challenge for the future development of agnostic tumor therapy. Understanding these diverse resistance mechanisms is crucial for developing strategies to overcome this challenge in cancer treatment.

Pembrolizumab and dostarlimab are both PD-L1 inhibitors that have been approved for *MSI-H/dMMR* solid tumors with poor efficacy seen in colorectal and pancreatic cancers. PD-L1 inhibition relies on antigen presentation for immune recognition (Figure 2). Thus, any alterations to the antigen presentation machinery, including loss of MHC expression, decrease the efficacy of these immune checkpoint inhibitors [42]. The level of neoantigen production may also contribute. For instance, some subpopulations of tumor cells may produce fewer neoantigens, which will result in decreased response rates [43].

The challenges surrounding the *NTRK* and *RET* fusion inhibitors are related to the rare incidence of these alterations and the diversity of fusion partners. *NTRK* and *RET* fusions are found in less than 5% of the population, with predominance in rare cancers. Thus, the approval of larotrectinib (n = 55), entrectinib (n = 54), selpercatinib (n = 41), and reprotrectinib (n = 88) were based on small patient populations. The most common *NTRK* fusion is *ETV6-NTRK3*. Demetri and colleagues found that patients receiving entrectinib had diverse fusions such as *ETV6-NTRK3* (n = 54, ORR = 74.1%), TPM3-*NTRK1* (n = 16, ORR = 56.3%), and *LMNA-NTRK1* (n = 6, ORR = 50%) [21]. Unique fusion partners are found in *RET* gene fusion-positive solid tumors. For instance, *CDCC6* and *NCOA4* are commonly found in papillary thyroid cancer. In contrast, *KIF5B* is most commonly found in NSCLC patients [44]. However, more research is required to elicit whether the fusion partner has a true effect on efficacy or if this is due to a low patient population.

Trastuzumab deruxtecan is a novel agnostic treatment; thus, more research is required to fully understand unique resistance mechanisms in pan-cancer cohorts. This may be due to co-mutations affecting *PIK3CA* and *PTEN*, which ultimately result in the hyperactivation of the *PI3K*/*AKT* pathway [45]. Some tumor cells may also be difficult to target due to lower *HER2* expression [46].

Resistance mechanisms also pose a challenge in clinical outcomes of tyrosine kinase inhibitors (TKIs) in targeting oncogenic drivers such as NTRK, RET, HER2, and BRAF. One well-characterized resistance mechanism involves gatekeeper mutations—amino acid substitutions at the target sites of TKIs—which reduce drug-target affinity, thereby diminishing therapeutic efficacy. Other studies of drug resistance reveal properties such as enhanced drug efflux, lysosomal sequestration, compensatory growth signals, and tumor microenvironment-driven immunosuppression, as well as metabolic and epigenetic alterations [47]. Bypass signaling through alternative pathways also plays a key role in resistance development [47].

NTRK fusion-positive cancers, specific mutations in the kinase domain, such as NTRK1 G595R and NTRK3 G623R, have been implicated in resistance by altering TKI binding affinity [48]. Similarly, in RET-targeted therapies, mutations such as RET V804M can lead to resistance, along with the activation of alternative signaling pathways, including MET and EGFR upregulation [49]. HER2-targeted therapies, like trastuzumab, face resistance mechanisms that include extracellular domain shedding of HER2, preventing effective antibody binding [50]. In BRAF-mutated cancers, resistance to BRAF inhibitors frequently involves reactivation of the MAPK pathway through either downstream mutations (e.g., MEK) or BRAF splice variants that promote dimerization, circumventing BRAF inhibition [51]. Resistance to immune checkpoint inhibitors (ICIs), such as PD-1/PD-L1 inhibitors, is similarly multifactorial. Tumor evasion strategies include loss or mutations in antigen-presentation mechanisms, such as reduced expression of major histocompatibility complex (MHC) molecules, limiting immune recognition. Tumors may also upregulate alternative immune checkpoints, such as TIM-3 or LAG-3, contributing to immune escape despite PD-1/PD-L1 blockade. Intrinsic tumor factors, such as mutations in the JAK/STAT pathway, disrupt interferon-gamma signaling, impairing immune responses [52]. Moreover, the tumor microenvironment fosters resistance by promoting infiltration of immunosuppressive cells, including regulatory T cells (Tregs) and myeloid-derived suppressor cells (MDSCs), which further inhibit effective anti-tumor immunity [53].

(b)Are there any diagnostic challenges?

Pembrolizumab was approved for solid tumors with *TMB-H* ≥ 10 mut/Mb; however, this cutoff value is still controversial. Different cancer histologies have different rates of neoantigen production. Thus, the TMB value may range between 0.001 mut/Mb to 1000 mut/Mb [54]. Thus, cancer types with a maximum TMB value of less than 10 mut/Mb may be missed despite deriving benefit from immunotherapy. Budczies and colleagues suggest that the cutoff point should be individualized for each cancer type [55]. Multiple technical issues such as panel size, adequate biopsy samples, and heterogeneous methods used in the literature also contribute to the diagnostic challenges in this population [56]. For instance, some studies utilize whole-genome sequencing (WGS), whereas other studies use small gene panels. TMB harmonization projects aim to establish a uniform TMB cutoff value and assay size that is optimal for pan-cancer cohorts [57].

The abundance and discordance of diagnostic tools must also be taken into consideration. For *NTRK* and *RET* fusions, multiple methods, such as fluorescence in situ hybridization (FISH), immunohistochemistry (IHC), and next-generation sequencing (NGS), are used. However, FISH and IHC may not be useful if the fusion partner is not known. In contrast, NGS conducts a more comprehensive analysis of the genome, which may detect multiple alterations, including unknown fusions. However, the feasibility and cost-effectiveness of this method must be considered for the clinical setting. The diagnostic recommendations depend on the cancer type. Solid tumors with higher *NTRK* and *RET* fusions are screened with FISH, followed by NGS confirmation. However, in cancers where fusions are uncommon, NGS may be the initial step [58]. Similarly, there may also be difficulties in diagnosing *HER2*-expressing solid tumors, especially in non-breast cancer patients. Currently, *HER2* status is determined by IHC, followed by FISH for equivocal cases (IHC2+). However, some studies have found discordance between the two methods [59]. Adding additional diagnostic tools such as NGS may aid in the diagnosis and treatment of *HER2*-positive solid tumors, especially in non-breast cancer patients [60].

(c)Does the tumor microenvironment (TME) play a role?

The tumor microenvironment (TME) is defined as the molecules, cells, and other non-cancerous components surrounding the tumor cells. The composition may vary between the different cancer types, with some contributing more to tumor growth, progression, and/or resistance. Critical members of the TME include stromal cells, immune cells, blood vessels, and the extracellular matrix [61]. The presence of cancer-associated fibroblasts (CAFs) creates a pro-oncogenic environment due to ECM modulation and secretion of pro-tumor factors [62]. Tumor-associated macrophages (TAMs), T cells, neutrophils, and other immune cells also influence the efficacy of targeted therapy. A proinflammatory TME stimulates cellular invasion and metastasis. In contrast, an immunosuppressive TME has been associated with unresponsiveness to immunotherapy [61]. The inflammatory environment also carries a regulatory role in angiogenesis. Furthermore, a hypoxic microenvironment may increase tumor angiogenesis, which promotes tumor growth [63]. The complex interaction between the tumor cells and the microenvironment may explain the heterogeneous response rates to tumor-agnostic therapy. However, the exact mechanism has not been fully elicited.

(d)Does tumor neoantigen burden (TNB) predict immunotherapy outcomes?

Other considerations include the role of tumor neoantigen burden (TNB) in relation to genomic instability. TNB is defined as the number of neoantigens per megabase in a given genome region. Increased genomic instability has been associated with a higher production of tumor neoantigens, which in turn may enhance T cell-specific immune responses [64]. While tumor mutational burden (TMB) has been used as a biomarker for predicting immunotherapy responses, it may not capture the full scope since not all mutations result in neoantigen production. Therefore, TNB has been proposed as a potentially more accurate biomarker [64]. Studies have shown that a high TNB correlates with improved progression-free survival and overall survival in patients with non-small cell lung cancer (NSCLC) treated with PD-1 inhibitors [65]. Similarly, in metastatic melanoma, patients receiving cytotoxic T-lymphocyte-associated protein 4 (CTLA-4) inhibitors demonstrated clinical benefit when their tumors exhibited a high TNB [66]. It is important to note that the current understanding of TNB highlights the involvement of other factors, such as the tumor microenvironment and specific genetic mutations, which contribute to therapy response. For example, tumor-infiltrating lymphocytes (TILs) are essential for neoantigen-driven immune responses, as increased TNB promotes T cell activation and proliferation, leading to greater TIL presence [64]. Future challenges include refining methods for predicting neoantigens and conducting further research to optimize their use in clinical settings.

(e)What agnostic therapies are on the horizon?

There are currently eight FDA-approved therapies, with new approvals likely on the way. Emerging targets include *KRAS*, *BRCA*, *FGFR*, *ALK*, *ROS1*, *VHL*, and *KIT*.

Sotorasib and adagrasib are promising *KRAS^G12C^* inhibitors, although evaluation of a pan-cancer cohort is in development. The CodeBreaK100 trial (phase I) evaluated 129 patients who received sotorasib monotherapy. The ORR was 32.2%, 7.1%, and 14.3% in patients with NSCLC, CRC, and other cancer types, respectively [67]. In contrast, adagrasib had a 35.1% response rate in *KRAS^G12C^*-positive solid tumors other than NSCLC and CRC [68].

Moreover, erdafitinib, the first pan-*FGFR* tyrosine kinase inhibitor, has also received FDA approval for the treatment of metastatic urothelial carcinoma in patients who have previously undergone therapy with a PD-1 or PD-L1 inhibitor. This approval was based on the results from the single-arm phase 2 RAGNAR study, which, at a median follow-up of 12.9 months, reported an objective response rate of 30% across 16 distinct tumor types. Notable among the grade 3 or higher treatment-emergent adverse events were stomatitis, palmar-plantar erythrodysesthesia syndrome, and hyperphosphatemia [69].

The emergence of T-Dxd has paved the way for the approval of antibody conjugate drug (ACD) in the future. Its ability to deliver a cytotoxic payload directly to cancer cells, while sparing healthy tissue, has showcased the potential of ADCs in improving treatment outcomes. The difficulty with ACD is that there can be significant effects on off target tissues, but T-Dxd’s safety profile has proven to be a milestone in that aspect.

The realm of tumor-agnostic drug development is advancing swiftly, demonstrating the importance of continuous post-marketing research to fully elucidate the safety profiles of these groundbreaking therapies. This ongoing evaluation is crucial for enhancing the therapeutic landscape of oncology and tailoring treatments to individual patient needs.

## 4. Conclusions

In conclusion, the domain of tumor-agnostic therapies holds considerable potential for the advancement of oncological care. The uptrend in both accelerated and full approvals of such drugs displays a promising trajectory toward efficacious treatments applicable across a diverse spectrum of malignancies. Ongoing clinical trials are instrumental in delineating the safety profiles of these novel interventions, emphasizing the necessity for meticulous evaluation of each therapeutic agent on a case-by-case basis, balancing its risks against its therapeutic benefits. This evolving landscape signals a paradigm shift in cancer treatment, moving toward more personalized and precision-based approaches.

## Figures and Tables

**Figure 1 cancers-16-03642-f001:**
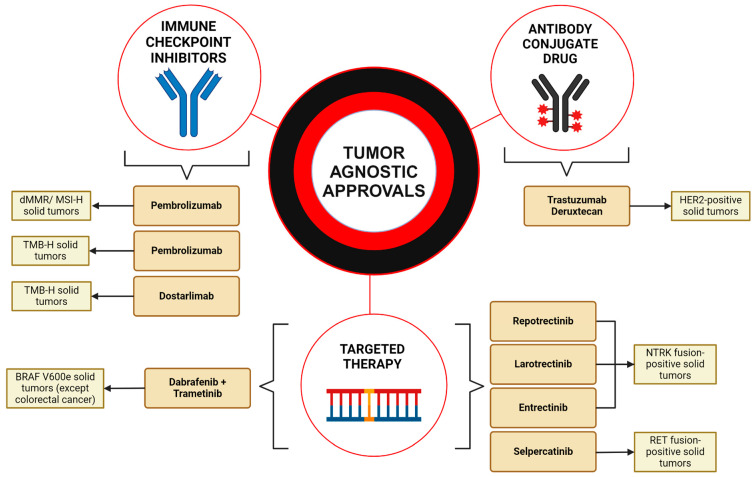
**Overview of the FDA-approved tumor-agnostic treatments.** This figure summarizes the tumor-agnostic treatments approved by the U.S. Food and Drug Administration (FDA). Immune checkpoint inhibitors include pembrolizumab, approved for both *dMMR/MSI-H* and *TMB-H* solid tumors, and dostarlimab, approved for *TMB-H* cancers. Targeted therapies include one combinational regimen consisting of dabrafenib plus trametinib, approved for *BRAF^V600E^* non-CRC tumors. Larotrectinib, entrectinib, and repotrectinib were both approved for *NTRK* fusions. Selpercatinib was approved for *RET* fusion-positive cancers. Trastuzumab deruxtecan represents the first antibody conjugate drug approved for *HER2*-positive cancers. These targeted therapies represent the emerging era of personalized cancer care driven by mutation type across multiple tumor types. Created in BioRender. Thein, K. (2024) BioRender.com/d44b340 (accessed on 12 March 2024).

**Figure 2 cancers-16-03642-f002:**
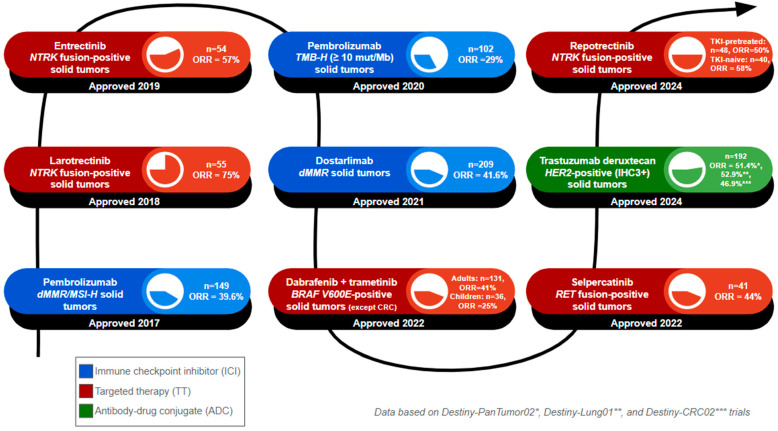
**Timeline of tumor-agnostic FDA approvals.** The number of FDA-approved tumor-agnostic therapies continues to grow, reflecting advances in targeted cancer treatments. This figure summarizes the year of FDA approval, the name of the approved regimen, and the targeted biomarker, and lists the patient population (n) and objective response rate (ORR) of the clinical trials that formed the basis of approval. 1. **Pembrolizumab** was approved in 2017 for *dMMR/MSI-H* solid tumors, based on 149 patients, with an ORR of 39.6%. 2. **Larotrectinib** was approved in 2018 for *NTRK* fusion-positive solid tumors, based on 55 patients, with an ORR of 75%. 3. **Entrectinib** was approved in 2019 for *NTRK* fusion-positive solid tumors, based on 55 patients, with an ORR of 75%. 4. **Pembrolizumab** was approved in 2020 for *TMB-H* solid tumors. The cutoff value used for *TMB-H* status was at least 10 mutations per megabase, based on 102 patients, with an ORR of 29%. 5. **Dostarlimab** was approved in 2021 for *dMMR* solid tumors, based on 209 patients, with an ORR of 41.6%. 6. **Dabrafenib plus trametinib** was approved in 2022 for *BRAF V600E* solid tumors (excluding colorectal cancers), based on 131 adult patients with an ORR of 41% and 36 pediatric patients with an ORR of 25%. 7. **Selpercatinib** was approved in 2022 for *RET* fusion-positive solid tumors, based on 41 patients, with an ORR of 44%. 8. **Trastuzumab deruxtecan** was approved in 2024 for *HER2*-positive (immunohistochemistry 3+ score) solid tumors, based on a pooled patient population of 192 across three key clinical trials: Destiny-PanTumor02 (ORR = 51.4%), Destiny-Lung01 (ORR = 52.9%), and Destiny-CRC02 (ORR = 46.9%). 9. **Repotrectinib** was approved in 2024 for *NTRK* fusion-positive solid tumors. Patients were divided into TRK tyrosine kinase inhibitor (TKI)-pretreated and TKI-naïve cohorts. The TKI-pretreated group, with 48 patients, had an ORR of 50%, while the TKI-nave cohort, with 40 patients, had an ORR of 58%.

**Figure 3 cancers-16-03642-f003:**
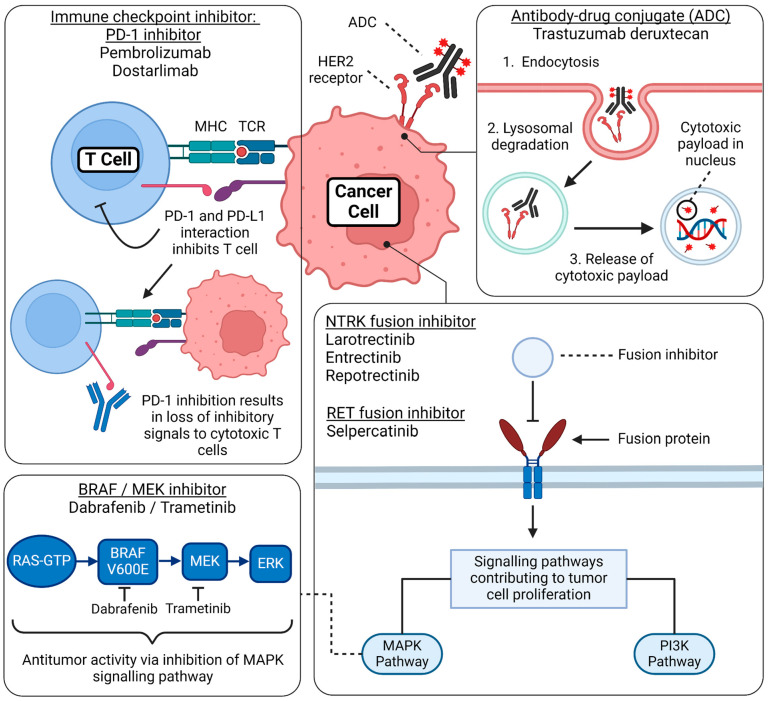
**Mechanism of action of FDA-approved tumor-agnostic therapy.** This figure summarizes the basic mechanism of action regarding the FDA-approved tumor-agnostic therapies. **Immune checkpoint inhibitor—PD-1 inhibitors:** PD-1 is a receptor expressed on T cells, and its ligand, PD-L1, is expressed on cancer cells. When PD-1 binds to PD-L1, it suppresses the activity of cytotoxic T cells, allowing tumor cells to evade immune detection and proliferate unchecked. PD-1 inhibitors, such as pembrolizumab and dostarlimab, block this interaction, restoring T-cell activity and enhancing the immune system’s ability to detect and destroy cancer cells. **Antibody-drug conjugate (ADC):** Trastuzumab deruxtecan consists of a monoclonal antibody, a cytotoxic payload, and a linker. Upon binding to the HER2 receptor on the cancer cell surface, the ADC is internalized through endocytosis. Inside the cell, the endosome fuses with a lysosome, where the complex is degraded, releasing the cytotoxic drug. This payload damages the tumor’s DNA, effectively inhibiting cell proliferation and inducing cancer cell death. **BRAF/MEK inhibitor combination:** Dabrafenib (a BRAF inhibitor) combined with trametinib (a MEK inhibitor) targets the MAPK signaling pathway, a critical cascade of phosphorylation events involving RAS, BRAF, MEK, and ERK proteins. This pathway regulates cell proliferation, survival, and differentiation. In cancers with BRAF V600E mutations, abnormal activation of this pathway leads to uncontrolled tumor cell growth. By blocking the mutant BRAF protein and inhibiting MEK downstream, this combination therapy effectively disrupts hyperproliferation and prevents paradoxical activation of the MAPK pathway. **NTRK fusion inhibitors and RET fusion inhibitors:** Gene fusions involving the NTRK and RET genes lead to the production of constitutively active receptor tyrosine kinases, which continuously signal even in the absence of normal regulatory inputs. These aberrantly activated receptors stimulate key proliferative and survival pathways, including the MAPK and PI3K/AKT pathways, driving oncogenic signaling that promotes tumor growth, survival, and metastasis. Targeted therapies, such as NTRK inhibitors (e.g., larotrectinib and entrectinib) and RET inhibitors (e.g., selpercatinib and pralsetinib), block these fusion-driven tyrosine kinases, effectively halting tumor progression by disrupting these essential growth signals. Created by BioRender. Thein, K. (2024) BioRender.com/i35k906 (accessed on 12 March 2024).

**Table 1 cancers-16-03642-t001:** Summary of FDA tumor-agnostic treatment.

Drug Name (s)	Approval Date (MM/DD/YYYY)	Indication	Clinical Trial (s)	Cohort (n)	Outcome Measures Used for Approval	Ref
Pembrolizumab	05/23/2017	*MSI-H/dMMR* solid tumors for adult and pediatric population	KEYNOTE-016,-164, -012, -028, and -158	149	ORR = 39.6% (95% CI: 31.7, 47.9), CR = 11, PR = 48, 78% with DOR ≥ 6 months	[3]
Larotrectinib	11/26/2018	*NTRK* fusion-positive solid tumors for adult and pediatric population	LOXO-TRK-1400, NAVIGATE, SCOUT	55	ORR = 75% (95% CI: 61, 85), CR = 22%, PR = 53%, mDOR = NR, 73% with DOR ≥ 6 months, 63% with DOR ≥ 9 months, 39% with DOR ≥ 12 months	[4]
Entrectinib	08/15/2019	*NTRK* fusion-positive solid tumors for adult and pediatric population (12 years or older)	ALKA, STARTRK-1, STARTRK02	54	ORR = 57% (95% CI: 43, 71), 68% with DOR ≥ 6 months, 45% with DOR ≥ 12 months	[5]
Pembrolizumab	06/15/2020	*TMB-H* (≥10 mut/mb) solid tumors for adult and pediatric population	KEYNOTE-158	102	ORR = 29% (95% CI: 21, 39), CR = 4%, PR = 25%, mDOR = NR, 57% with DOR ≥ 12 months. 50% with DOR ≥ 24 months	[6]
Dostarlimab	08/17/2021	*MSI-H/dMMR* solid tumors for adult population	GARNET	209	ORR = 41.6% (95% CI: 34.9, 48.6), CR = 9.1%, PR = 32.5%, mDOR = 34.7 (2.6 to 35.8+) months, 95.4% with DOR ≥ 6 months	[7]
Dabrafenib plus trametinib	06/22/2022	BRAF V600E solid tumors * for patients 6 years or older	BRF117019, NCI-MATCH, CTMT212X2101	Adults: 131 Children: 36	Adults: ORR = 41% (95% CI: 33, 50) Children: ORR = 25% (95% CI: 12, 42), 78% with DOR ≥ 6 months, 44% with DOR ≥ 24 months	[8]
Selpercatinib	09/21/2022	RET fusion-positive solid tumors for adult population	LIBRETTO-001	41	ORR = 44% (95% CI: 28, 60), mDOR = 24.5 months (95% CI: 9.2, NE), 67% with DOR ≥ 6 months	[9]
Trastuzumab Deruxtecan	04/05/2024	HER2-positive (IHC3+) solid tumors for adult population	1. DESTINY-PanTumor02 2. DESTINY-Lung01, 3. DESTINY-CRC02	192	1. ORR = 51.4% (95% CI: 41.7, 61.0), mDOR = 19.4 (1.3 to 27.9+) months 2. ORR = 52.9% (95% CI: 27.8, 77.0), mDOR = 6.9 (4.0 to 11.7+) months 3. ORR = 46.9% (95% CI: 34.3, 59.8), mDOR = 5.5 (1.3+ to 9.7+) months	[10]
Reprotrectinib	06/13/2024	NTRK gene fusion-positive solid tumors for adult and pediatric patients 12 years or older	TRIDENT-1	88	TKI naïve: ORR = 58% (95% CI: 41,73), mDOR = not estimable (NE) TKI-pretreated: 50% (95% CI: 35,65), mDOR = 9.9 (7.3, 13.0) months	[11]

* except colorectal cancer. Abbreviations: BRAF, V-Raf Murine Sarcoma Viral Oncogene Homolog B; CR, complete response; dMMR, deficient mismatch repair; DOR, duration of response; IHC, immunohistochemistry; mDOR, median duration of response; MSI-H, cMicrosatellite instability High; mut/mB, mutations per megabase; NE, not estimable; NTRK, neurotrophic tropomyosin-receptor kinase; ORR, objective response rate; PR, partial response; RET, rearranged during transfection; TMH-H, tumor mutational burden-high.

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
