# Peer review of "Histology Agnostic Drug Development: An Updated Review"

_cancers, 2024, doi:10.3390/cancers16213642_

Round 1

Reviewer 1 Report

Comments and Suggestions for Authors

The article entitled “Histology Agnostic Drug Development: An Updated Review” has been evaluated. This study focused on the recent advancements in precision oncology, including FDA approvals for several targeted therapies, including trastuzumab, selpercatinib, pembrolizumab, larotrectinib, entrectinib, dostarlimab, and reprotectinib. However, there are several critical points to address in the manuscript.

1. It is crucial to ensure that the article is not only informative but also engaging, aiming to capture the interest of the general audience of the journal.

2. Enhancing the presentation of figures to make them more appealing and understandable would greatly contribute to bolstering the overall impact and influence of the journal. The objective is to motivate readers to interact with the content and gain valuable insights, irrespective of their level of expertise in the subject area.

3. The abstract section should be expanded and elaborated upon in a more detailed manner to provide a comprehensive overview of the study's findings and significance.

4. The introduction section should be rewritten with a stronger emphasis on histology Agnostic Drug Development to effectively set the stage for the rest of the manuscript.

5. Additionally, it is recommended that the authors add a new section after the introduction, focusing on "Histology-Agnostic Treatments: Promises and Pitfalls."

6. Furthermore, a discussion about trial designs for histology-agnostic drug development should be incorporated to provide a broader understanding of the topic.

Comments on the Quality of English Language

Minor editing of English language required.

Author Response

Reviewer Comments:

The article entitled “Histology Agnostic Drug Development: An Updated Review” has been evaluated. This study focused on the recent advancements in precision oncology, including FDA approvals for several targeted therapies, including trastuzumab, selpercatinib, pembrolizumab, larotrectinib, entrectinib, dostarlimab, and reprotectinib. However, there are several critical points to address in the manuscript.

  1. It is crucial to ensure that the article is not only informative but also engaging, aiming to capture the interest of the general audience of the journal.

Author Response: Thank you for the comment and we have changed the language throughout the manuscript accordingly.

  1. Enhancing the presentation of figures to make them more appealing and understandable would greatly contribute to bolstering the overall impact and influence of the journal. The objective is to motivate readers to interact with the content and gain valuable insights, irrespective of their level of expertise in the subject area.

Author Response: Thank you for the comment. We have added figures about mechanism as well as a a timeline of approvals of drugs.

  1. The abstract section should be expanded and elaborated upon in a more detailed manner to provide a comprehensive overview of the study's findings and significance.

Author’s Response: Thank you for the comment, we have expanded upon the abstract.

“Recent advancements in oncology have led to the development of histology-agnostic therapies, which target genetic alterations irrespective of the tumor’s tissue of origin. This review aims to provide a comprehensive update on the current state of histology-agnostic drug development, focusing on key therapies, including pembrolizumab, larotrectinib, entrectinib, dostarlimab, dabrafenib plus trametinib, selpercatinib, trastuzumab deruxtecan, and reprotrectinib. We performed a detailed analysis of each therapy’s mechanism of action, clinical trial outcomes, and associated biomarkers. The review further explores challenges in drug resistance, such as adaptive signaling pathways and neoantigen variability, as well as diagnostic limitations in identifying optimal patient populations. While these therapies have demonstrated efficacy in various malignancies, significant hurdles remain, including intratumoral heterogeneity and resistance mechanisms that diminish treatment effectiveness. We propose considerations for refining trial designs and emerging biomarkers, such as tumor neoantigen burden, to enhance patient selection. These findings illustrate the transformative potential of histology-agnostic therapies in precision oncology but highlight the need for continued research to optimize their use and overcome existing barriers.”

  1. The introduction section should be rewritten with a stronger emphasis on histology Agnostic Drug Development to effectively set the stage for the rest of the manuscript.

Author’s response: Thank you for the comment, we have edited the introduction and added a simple summary.

  1. Additionally, it is recommended that the authors add a new section after the introduction, focusing on "Histology-Agnostic Treatments: Promises and Pitfalls."

Author Response: Thank you for the comment. We have added a section focusing on the promises and pitfalls of histology-agnostic treatment.

“Histology-agnostic therapies have revolutionized cancer treatment by targeting specific molecular alterations irrespective of the tumor’s tissue of origin. These therapies, such as larotrectinib and entrectinib for NTRK fusions and pembrolizumab for tumors with MSI-H, have shown promising results in a variety of malignancies, demonstrating the potential for broad application. However, the clinical utility of these treatments is limited by several key challenges. Heterogeneous responses across different tumor types remain a major issue, as exemplified by the diminished efficacy of BRAF inhibitors in colorectal cancer compared to melanoma due to feedback activation of alternative pathways. Resistance mechanisms, including secondary mutations, gene amplifications, and adaptive pathway reactivation, further complicate treatment and can lead to therapeutic failure over time. Understanding the various pharmacologic mechanisms of action is also a key aspect in understanding the efficacy and resistance of these treatments (figure 3). Additionally, effective implementation requires advanced diagnostic testing, such as comprehensive genomic profiling, which may not be widely available, particularly in low-resource settings. Patient stratification is another hurdle, as identifying the small subset of patients harboring these rare mutations can complicate trial recruitment and reduce the statistical power of studies. Addressing these barriers through improved trial designs, better understanding of resistance mechanisms, and the development of standardized diagnostic criteria is essential to maximize the impact of histology-agnostic therapies and broaden their clinical application​.”

  1. Furthermore, a discussion about trial designs for histology-agnostic drug development should be incorporated to provide a broader understanding of the topic.

Author Response: Thank you for the comment. However, in this review, we provide an update on these therapies, focusing on several newly developed drugs that are changing the way cancer is treated. Our goal is to explain how these drugs work, the benefits they offer, and the challenges that remain, such as resistance to treatment. The trial designs for histology-agnostic drug development is a whole separate new chapter which we’re not covering in this already extensive review.

Reviewer 2 Report

Comments and Suggestions for Authors

The authors report histology agnostic drug development.

1.     The authors should illustrate the mechanism of action of the drug and intracellular signaling in the Figure.

Author Response

The authors report histology-agnostic drug development.

Comment #1:  The authors should illustrate the mechanism of action of the drug and intracellular signaling in the Figure.

Author’s Response: Thank you for the recommendation, we have added an illustration describing the mechanisms of the different therapies.

Reviewer 3 Report

Comments and Suggestions for Authors

Nice review work. Comprehensive and clear.

Just for readers interest, may I suggest adding:

a. A Figure or cartoon describing the pathway & mechanism of action of at least the most widely used class - ICI? The different targets of ICI may be helpful for understanding ......

Author Response

Nice review work. Comprehensive and clear.

Comment #1: Just for readers' interest, may I suggest adding: A Figure or cartoon describing the pathway & mechanism of action of at least the most widely used class - ICI? The different targets of ICI may be helpful for understanding ......

Author’s Response: Thank you for your kind comment. We have added a figure to illustrate the mechanisms of different therapies.

Reviewer 4 Report

Comments and Suggestions for Authors

The manuscript with the title “Histology Agnostic Drug Development: An Updated Review” is well organized but I will give some suggestion to the authors:

1)      A minor editing of English language required

2)      Authors mentioned MSI and TMB as the only biomarkers  that used for Pembrolizumab. What is applied about other cancer type as NSCLC, which markers is used in that type of cancer?

3)      A section about the resistance mechanism against immunotherapy and TKIs is required?

4)      I strongly suggest to add a section about neo-antigen as a putative biomarker for immunotherapy (TNB: tumor neoantigen burden).

5)      One or more figure which explain the main way of action of immunotherapy, and the others agents.  

Comments on the Quality of English Language

-

Author Response

Reviewer Comments:

The manuscript with the title “Histology Agnostic Drug Development: An Updated Review” is well organized but I will give some suggestion to the authors:

1)      A minor editing of English language required

Author Response: Thank you for your comment. We have changed the language of the manuscript accordingly.

2)      Authors mentioned MSI and TMB as the only biomarkers  that used for Pembrolizumab. What is applied about other cancer type as NSCLC, which markers is used in that type of cancer?

Author Response: Thank you for your comment. As this is the topic of Histology agnostic drug development, we don’t cover other mutations/ alterations specific to the tumor type. However, we cover the title of -

“What agnostic therapies are on the horizon?

There are currently 8 FDA approved therapies, with new approvals likely on the way. Emerging targets include KRAS, BRCA, FGFR, ALK, ROS1, VHL, and KIT. …etc…”

3)    A section about the resistance mechanism against immunotherapy and TKIs is required?

Author Response:  Thank you for your comment, we have added a section about resistance mechanisms

“Resistance mechanisms also pose a challenge in clinical outcomes of tyrosine kinase inhibitors (TKIs) in targeting oncogenic drivers such as NTRK, RET, HER2, and BRAF. One well-characterized resistance mechanism involves gatekeeper mutations—amino acid substitutions at the target sites of TKIs—that reduce drug-target affinity, thereby diminishing therapeutic efficacy. Other studies of drug resistance reveal properties such as enhanced drug efflux, lysosomal sequestration, compensatory growth signals, tumor microenvironment-driven immunosuppression, as well as metabolic and epigenetic alterations. Bypass signaling through alternative pathways also plays a key role in resistance development.

 NTRK fusion-positive cancers, specific mutations in the kinase domain, such as NTRK1 G595R and NTRK3 G623R, have been implicated in resistance by altering TKI binding affinity. Similarly, in RET-targeted therapies, mutations such as RET V804M can lead to resistance, along with activation of alternative signaling pathways, including MET and EGFR upregulation.HER2-targeted therapies, like trastuzumab, face resistance mechanisms that include extracellular domain shedding of HER2, preventing effective antibody binding. In BRAF-mutated cancers, resistance to BRAF inhibitors frequently involves reactivation of the MAPK pathway, through either downstream mutations (e.g., MEK) or BRAF splice variants that promote dimerization, circumventing BRAF inhibition.Resistance to immune checkpoint inhibitors (ICIs), such as PD-1/PD-L1 inhibitors, is similarly multifactorial. Tumor evasion strategies include loss or mutations in antigen-presentation mechanisms, such as reduced expression of major histocompatibility complex (MHC) molecules, limiting immune recognition. Tumors may also upregulate alternative immune checkpoints, such as TIM-3 or LAG-3, contributing to immune escape despite PD-1/PD-L1 blockade. Intrinsic tumor factors, such as mutations in the JAK/STAT pathway, disrupt interferon-gamma signaling, impairing immune responses. Moreover, the tumor microenvironment fosters resistance by promoting infiltration of immunosuppressive cells, including regulatory T cells (Tregs) and myeloid-derived suppressor cells (MDSCs), which further inhibit effective antitumor immunity.”

4)      I strongly suggest to add a section about neo-antigen as a putative biomarker for immunotherapy (TNB: tumor neoantigen burden).

Author Response: Thank you for your comment, we have added a section about neo-antigens.

5)      One or more figure which explain the main way of action of immunotherapy, and the others agents. 

Author’s Response: Thank you for your comment. We have added a figure to illustrate the mechanisms of different therapies.

Round 2

Reviewer 1 Report

Comments and Suggestions for Authors

The authors made significant revisions based on the reviewer's comments, and the manuscript can be acceptable for publication.

Comments on the Quality of English Language

Minor editing of English language required.

Reviewer 2 Report

Comments and Suggestions for Authors

none